# The Relationship between Self-Compassion and Sleep Quality: An Overview of a Seven-Year German Research Program

**DOI:** 10.3390/bs10030064

**Published:** 2020-03-06

**Authors:** Sebastian Butz, Dagmar Stahlberg

**Affiliations:** Faculty of Social Sciences, University of Mannheim, 68159 Mannheim, Germany; sbutz@mail.uni-mannheim.de

**Keywords:** self-compassion, sleep quality

## Abstract

Poor sleep quality is highly prevalent in modern societies and negatively linked to various health outcomes. While previous research has demonstrated preliminary evidence for self-compassion as a tool for improving sleep quality, this review provides a meta-analysis of respective published and unpublished results of our own research group using German samples. A total of nine studies are included (*N* = 956 participants), consisting of both correlational and experimental data. Across these studies, there was a medium correlation between self-compassion and subjective sleep quality, *r* = 0.303, 95% CI (0.244; 0.360). In three experimental studies, a small increase in participants’ self-reported sleep quality emerged, in comparison to control conditions, Hedges’ s *g* = 0.484, 95% CI (0.148; 0.821). Limitations on study level concern both the small sample sizes and short-term analyses of intervention effects. As a conclusion, this review supports both the correlational and causal relationship between self-compassion and increased subjective sleep quality across diverse operationalizations and samples. Future research should focus on the moderating effects of intervention type, duration of intervention effects, and type of target population.

## 1. Introduction

In modern societies, poor sleep quality is highly prevalent—depending on the insomnia definition, estimates range from 6% (insomnia diagnosis as defined by the DSM-IV) to 33% (insomnia symptoms) in the general population [1]. Due to the heterogeneity of insomnia definitions, we use the term poor sleep quality in the given work to refer to self-reported insomnia symptoms with varying degrees of severity (e.g., having trouble to fall asleep, difficulties sleeping through the night, waking up tired in the morning). The prevalence of poor sleep is noteworthy because it negatively affects an individual’s overall quality of life beyond the nocturnal effects [2]. For instance, insomnia symptoms predict lower mental health, including depression, in longitudinal epidemiological studies [3]. Moreover, sleep disorders are linked to negative work outcomes, including occupational accidents, absenteeism, and presenteeism [4]. In addition, people suffering from insomnia symptoms report a significant impairment of daytime functioning with regard to several social, emotional, and physical domains [5], even after controlling for comorbid medical illnesses. They also report an overall lower quality of life comparable to that of individuals with chronic medical conditions [6].

Based on these correlates of poor sleep, it is also worrisome that previous research has shown that the number of people diagnosed with chronic insomnia is increasing year by year in the United States [7]. Presently, a substantial part of the population does not seem to achieve the recommended sleep duration of between seven and eight hours [8], and one study even shows a gradual decrease of more than one hour in sleep time from 1905 to 2008, both in children and in adolescents [9]. Poor sleep quality is not only prevalent and steadily increasing, but also the associated economic costs are high. It is estimated that people diagnosed with insomnia in the United States cause 60% higher health care costs compared to people without a disease [7], with annual costs for the health care system between 92 to 107 billion dollars. A meta-analysis of large representative surveys involving more than 600,000 adults and more than 30,000 children showed that both children and adults with short sleep periods (<6 h per night) had a higher chance of suffering from obesity resulting in individual suffering but also further costs for the health care system and society at large (odds ratio, OR, children = 1.89, OR adults = 1.55) compared to persons without sleep deprivation [10].

As a solution, experts and guidelines recommend cognitive behavioral therapy for insomnia (CBT-I) and short-term pharmacological treatment [11,12]. Although CBT-I is effective with medium to large effect sizes [13], the threshold to search for this kind of professional help seems high. According to a survey (*N* = 432) from a German health insurance company (DAK) in 2016, about 70% of employees who met the DSM-IV criteria for insomnia have never consulted their physician because of sleep problems, and those who did only sought medical treatment after a mean delay of two years [14]. This can cause further problems; when not actively treated, the sleep quality of chronic insomniacs tends to deteriorate. Representative survey data revealed that severe insomnia lasts for a median of four years [15], and 88.2% of patients continue to report sleep disturbances five years after the onset of the disorder [16]. Furthermore, one study found that only 56% of individuals report symptom remission after 10 years [17].

Why do people abstain from seeking professional help for their sleep problems? One answer may be that people (erroneously) perceive sleeping disorders as something they may tackle themselves successfully. According to one study, four out of 10 people with insomnia said they had “helped themselves” either by using non-prescription medication or by drinking alcohol [18]. Another answer may be that people do not know about professional treatment options or that they perceive the abovementioned treatments as too costly in terms of money or time to be invested. Thus, low-cost treatments that can function as both supplements to standard treatment and as preemptive measures to prevent a chronic status of insomnia are clearly needed.

There is evidence that mindfulness-based approaches (e.g., mindful meditation) serve as an auxiliary treatment toward higher sleep quality [19]. While traditional mindfulness-based therapies do not differ from short-term cognitive-behavioral therapies in terms of time expenses (>8 sessions) and with regard to decreases in various psychological symptoms, including anxiety disorders, depression, and stress experiences [20], shorter mindfulness interventions (≤8 sessions) are particularly suitable for non-clinical samples [21]. For example, in a randomized controlled intervention study, employees at a Danish bank received 8 weeks of mindfulness exercises (60 min/week) that led to an increase of subjective sleep quality [22]. Thus, mindfulness-based approaches could prevent chronification of poor subjective sleep quality via short interventions.

The present paper introduces another concept closely related to mindfulness that can make people less vulnerable to sleep problems and may even help them to overcome these problems—self-compassion. The construct of self-compassion as introduced by Neff is defined as a positive and caring attitude of a person toward her- or himself in the face of failures and individual shortcomings [23]. Accordingly, self-compassion can serve as an inner attitude that results in people reacting with empathy to negative events in their life that can occur with or without their own responsibility. Self-compassion is based on the motivation a) to pay close attention to one’s own suffering and b) to reduce one’s own suffering. The construct of self-compassion can be further divided into six different subcomponents: Three positive aspects (self-kindness, common humanity and mindfulness) stand diametrically opposed to the associated negative aspects (self-judgment, isolation, over-identification). Applied to the context of poor sleep, the subcomponents can be described as follows:Self-kindness vs. self-judgment. Self-compassionate people react with self-directed empathy and warmth when faced with daily stressors or the sleep problems themselves (instead of harshly criticizing themselves);Common humanity vs. isolation. Self-compassionate people cognitively classify their predicament as part of a universal human experience (instead of looking at themselves in isolation from others);Mindfulness vs. over-identification. Self-compassionate people pay attention to the suffering they have experienced (instead of avoiding the emotion or being totally carried away by it). 

Self-compassion is suggested as an emotion-regulating coping strategy in dealing with psychosocial stressors via a number of processes (i.e., emotional acceptance, self-distance, cognitive reassessments, reinforcement of positive emotions, and reduction of negative emotions) [24,25]. These adaptive effects of a high level of self-compassion can be explained with two theories: the broaden-and-build theory of positive emotions and the social mentality theory [26,27]. Based on broaden-and-build theory, self-compassion should foster persistent positive emotions that promote the development of resilience and well-being in an upward spiral; a person who develops a self-compassionate attitude should be more open to personal experiences, especially to negative emotions such as guilt, fear, helplessness. The postulated adaptive handling of negative emotions or self-discrepancies strengthens the development of a balanced perspective on one’s own weaknesses. This balance should in turn go hand in hand with positive emotions and increase one’s well-being. Social mentality theory postulates a biopsychosocial model for the explanation and prediction of adaptive effects and correlations of self-compassion, taking into account assumptions from evolution and attachment theory as well as neuroscience. The central assumption of social mentality theory regarding self-compassion is that self-compassion activates a soothing mindset focused on caring for the self that is associated with feelings of being safe and contend. On a physiological level, this mindset seems to be linked to neuro-endocrinological processes and the activation of the parasympathetic nervous system. For example, participants who visualized compassion for others and for themselves showed a decrease in heart rate variability and cortisol levels compared to participants who viewed neutral or relaxing pictures [28]. Thus, self-compassion not only seems to go hand in hand with positive emotions but also induces a soothing effect with regard to somatic arousal indicators such as heart rate variability.

To understand why and how self-compassion should predict a higher subjective sleep quality, it is important to connect the aforementioned self-compassion literature with etiological models explaining poor subjective sleep quality. Converging evidence from cognitive, endocrine, neurological, and behavioral domains provide clear evidence for hyperarousal as a key factor in insomnia [29]. Other work discusses related variables such as anxiety sensitivity, dysfunctional beliefs, and neuroticism [30]. Theoretically, a self-compassionate attitude should buffer against these key variables, as it reduces physiological arousal as well as processes maintaining hyperarousal long after the actual stressor has vanished (e.g., rumination). Recent empirical evidence stemming from cross-sectional data further corroborates the link between self-compassion and subjective sleep quality [31,32,33]. Furthermore, previous meta-analytical work has consistently confirmed positive correlations between self-compassion and mental health, for example in self-reports on life satisfaction, well-being, self-confidence, optimism, curiosity, gratitude, and social involvement (r = 0.47) [34]. Another meta-analysis showed a robust negative relationship between self-compassion and symptoms of general psychopathology, such as fear, depression, stress, rumination, thought suppression, perfectionism, and shame (r = −0.54) [35]. However, to date, there is no meta-analysis of the relationship between self-compassion and subjective sleep quality.

The primary goal of this meta-analysis on our own research in Germany is to gain a more reliable quantitative estimate of (a) the relationship between trait self-compassion and subjective sleep quality and (b) the efficacy of self-compassion interventions on subjective sleep quality. In our initial publication, self-compassion interventions resulted in an increase of self-reported sleep quality, compared to control conditions, in two independent data sets using an experimental design [36]. In the context of dealing with the current replication crisis in psychological research, many authors have asked to conduct an internal meta-analysis within one’s manuscript [37]. Thus, in this internal meta-analysis, we aim to expand these preliminary findings by providing an overview of a research program that we conducted from 2012 to 2019 on the link between self-compassion and subjective sleep quality. 

## 2. Materials and Methods

This internal meta-analysis differs from a systematic review in important ways because it only includes nine studies that we conducted in our lab group in Germany. Thus, this meta-analysis does not follow the protocol of a systematic review (e.g., there was no pre-registration and no electronic database search). The nine studies that were incorporated in the meta-analysis varied in several design features (e.g., online survey vs. paper–pencil questionnaire; correlational vs. experimental, between- and within-participants), and participants were recruited from different populations (i.e., students vs. patients with major depressive disorder). In addition, the study entries of this meta-analysis stem from unpublished data (see Table 1), with the exception of three independent data sets included in one publication [36]. Data on the year, type of publication, sample size, study design (cross-sectional vs. intervention), participant characteristics (study setting, proportion female, and mean age), and outcome measures are reported for all studies.

Self-compassion: Self-compassion was assessed using either the long (26 items) or short form (12 items) of the Self-Compassion Scale (SCS; SCS-SF) [38,39]. Both versions consist of items measuring three positive and three negative subcomponents of self-compassion (e.g., for isolation, “When I’m feeling down, I tend to feel like most other people are probably happier than I am”; for common humanity, “When things are going badly for me, I see the difficulties as part of life that everyone goes through”), with responses ranging from 1 = never to 5 = always. As the samples were recruited in Germany, all studies assessed self-compassion using a validated German translation [40]. In all studies, a total self-compassion score was computed (the negative subscale items were reversed before calculating subscale means), with higher scores representing a higher level of self-compassion.

Self-compassion interventions: Participants were randomly assigned to either a self-compassion intervention or control group. The self-compassion interventions differed across the three experimental studies; we used a self-compassionate loving-kindness meditation (LKM) [41] or writing prompts to activate or train the positive subcomponents of self-compassion [36,42]. In one study (ix), the expectation of treatment effects was also manipulated (positive vs. no positive expectation: “Previous studies show no vs. a short-term improvement in sleep quality through the following exercises”) to account for potential placebo effects. The treatment intensity ranged from 20 min in one session (study v) to a total of 50 min over the course of six sessions within one week (study ix). All self-compassion interventions followed an individual delivery method, either via paper–pencil (study vii) or online (study v, ix). Preceding every self-compassion intervention, participants had to reflect for three minutes on a personal problem that would constitute the object to which the self-compassion exercises would apply. The exact instructions read as follows:

“And now I’d like you to bring to mind some aspect of your personality, such as a mistake you’ve made, a failure that has been bothering you lately, something that you perhaps have been criticizing yourself for and that has made you feel inadequate in some way. Whatever this trait or action is, try to get in touch with your feelings about it. What does it make you feel like?”

Control groups were waiting groups with no additional instructions (with the exception of study v, where participants additionally reflected on a personal problem for three minutes to establish a baseline of ruminative thoughts in a student sample).

Manipulation checks were used in study v and ix. In study v, all participants had to describe their current emotional state using 13 self-compassionate adjectives directly after the intervention (or after no further instruction in the control condition), e.g., balanced, isolated, kind and loving, connected, ranging from 1 = absolutely disagree to 5 = totally agree. In study ix, we used a six item version (each item related to a self-compassion subcomponent, e.g., “I tried to accept my mistakes and weaknesses” for self-kindness, “I tried not to judge my thoughts” for mindfulness or “I tried to imagine other people who have similar thoughts” for common humanity), with the same response format as mentioned above.

Subjective sleep quality: Sleep quality was also assessed as a total score summing up different features of self-reported sleep quality. Such an overall score is most likely to provide a reliable and valid estimate of a person’s sleep quality [43] (p. 319). In six studies (i–iii; v–vi; ix), an index of sleep quality was used [43], comprising different sleep domains (e.g., “Did you experience problems falling asleep?”, with response scales ranging from 1 = not at all to 5 = very much). In one study (viii), sleep quality was assessed using the Pittsburgh Sleep Quality Index [44]. A total of 19 items comprise seven components (e.g., difficulties sleeping through the night), each of which has a range of 0 to 3 points, with 0 indicating no difficulty, whereas a score of 3 indicates severe difficulty. In two studies (iv and vi), the Insomnia Severity Index was applied (ISI) [45]. This index consists of five items (e.g., the severity of one’s problem with waking up too early), with response scales ranging from 0 = none to 4 = very severe. The time frame for the retrospective judgments of one’s sleep quality domains ranged from the last four weeks to last night (usually asked the morning after waking up).

## 3. Results

This review’s effect measures include Pearson’s correlation coefficients and Hedges’ *g*. As measures of consistency, Cochran’s Q was calculated. No additional analyses were performed. We conducted the meta-analysis across nine studies with a combined sample size of 956 participants with Comprehensive Meta-Analysis 3.3 and specified a random effects model. Across studies, participants’ mean age was 29.46 years, with a high percentage of female participants across studies (69.4%).

Correlational findings: Trait self-compassion was significantly related to self-reported overall sleep quality across the nine studies, *r* = 0.303, 95% CI (0.244; 0.360), *p* < 0.001 (see Figure 1). This represents a medium effect size [46]. The Q statistic indicated no significant heterogeneity in the effects, *Q*(8) = 2.48, *p* = 0.96.

Experimental Findings: Across the three experimental studies, a small increase of participants’ self-reported sleep quality emerged in comparison to passive control conditions, *g* = 0.484, 95% CI (0.148; 0.821), *p* = 0.005 (see Figure 2). Again, the Q statistic indicated no significant heterogeneity in the effects, *Q*(4) =0.664, *p* = 0.95.

A disadvantage of this analytical approach is that the different measurements are considered independent groups [47]. Therefore, we considered an alternative effect size calculation (Cohens’ *d*) to account for the hypothesized dependencies between measurements (i.e., we expected an increase of subjective sleep quality in the self-compassion interventions between pretest and posttest). This approach focuses only on the increase of sleep quality in self-compassion interventions in pre–post designs. In line with our hypothesis, across two studies and three separate self-compassion interventions, there was a small increase of participants’ subjective sleep quality between pretest to posttest, *d* = 0.368, 95% CI (0.232; 0.505), *p* < 0.001.

## 4. Discussion

This meta-analysis included nine studies using both correlational and experimental data, involving 956 participants (with one clinical sample). We found a medium correlation between self-compassion and subjective sleep quality (*r* = 0.303; can be converted to *d* = 0.636). The present review is limited because our review only focused on our own research and did not include previously published correlational research articles [31,32,33]. However, the previously reported range of the SC–SQ correlations (*r*s = 0.23 to 0.40) is comparable to the effect size in this meta-analysis. With regard to the three intervention studies, there was a small increase of participants’ sleep quality due to the self-compassion interventions, Hedges’ *g* = 0.484, 95% CI (0.148; 0.821). The effect sizes were controlled for pretest differences and reflect a comparison to passive control conditions. This is obviously a small database, and any drawn conclusions should keep this in mind.

One limitation of these results concerns the interpretation of the causal role of self-compassion on sleep quality. First, the three experimental studies investigated only short-term intervention effects, with a total exposition to the self-compassion material shorter than one hour. In comparison, Neff and Germer developed an eight-week training course based on mindfulness-based approaches (mindful self-compassion; MSC), where changes in self-compassion caused by the training compared to the pre-test also remained in a follow-up survey that took place a year later [48]. The increase in self-compassion was also related to the level of training—the more days participants meditated and the more hours a day they were spent on the self-compassion exercises, the higher the increase in self-compassion. Thus, while less than one hour of self-compassion exercises per week seems sufficient to increase participants’ subjective sleep quality, previous work suggests that more exposition might increase intervention effect sizes and sustainability of change with regard to sleep quality. However, if subjective sleep quality further improves with more time-consuming self-compassion interventions, these interventions would lose their asset of being efficient short-term approaches. Even then, given the reported effects, self-compassion training modules may also be useful in a setting where they are supplementary/complimentary to the CBT-I. Second, we tested only the causal route of self-compassion influencing sleep quality but not vice versa. While there is preliminary evidence for enhanced cognitive functioning and a more positive mood after a good night‘s sleep [49], future research should investigate whether and how sleep quality may also foster a self-compassionate attitude. Third, a potential explanation of the association between self-compassion and self-reported sleep quality could be that more self-compassionate people tend to look at their perceived sleep quality in a more positive way. However, in study iv, the correlations between self-compassion and sleep quality did not differ significantly regarding the order of presentation; thus, this relationship does at least not seem to depend on a heightened salience of either variable.

Another limitation of the present operationalization is the focus on self-reports (e.g., participants should subjectively assess their sleep quality in each study in a retrospective survey one hour after waking up). There are also problems with capturing self-compassion through self-reports: Various papers critically discuss the use of psychometric scales to measure constructs such as mindfulness and self-compassion, since these scales assess the respondents’ subjective perception rather than their actual expression [50]. The question of the validity to assess sleep quality via self-reports can be criticized for the same reasons. Other measures such as additional data from partners or other people close to the respective study participant as well as objective measures in the case of sleep quality may enhance data quality by reducing response distortions. However, regarding sleep quality, the objective measures do not only offer advantages over subjective measurement methods. For example, sleep quality changes already as a result of the recording in the sleep laboratory, since people outside of their usual sleeping environment sleep more restlessly on average [51]. Recording sleep quality using actigraphy also has several disadvantages: The validity of the method is controversial insofar as actigraphy only measures movement intervals. While movement intervals can highly correlate with sleep quality, actigraphy detects sleep disorders less reliably, especially in clinical populations [52]. For instance, nighttime television viewing can be incorrectly encoded as sleep time. To avoid such mistakes, researchers have to gather additional subjective information from the test subjects, which reduces the economics of the procedure. Most importantly, we would argue that because people with a chronically low quality of sleep differ from healthy sleepers especially regarding their subjective experience [53], future studies should continue to focus on an intervention-related increase in the subjective experience of sleep quality. In addition, future research should investigate whether self-compassion interventions can also lead to an increase in sleep quality using objective measurement methods.

Further limitations of the presented findings were that we used convenience sampling in all studies (six unpublished data sets were collected in partial fulfillment of the bachelor’s/master’s/dissertation theses at the University of Mannheim). Thus, it is possible that effect sizes will increase with larger sample sizes. Future research should systematically investigate the efficacy of different self-compassion interventions—ideally with a focus on a person–intervention fit (e.g., via introduction of a factor self-selection vs. random assignment). In addition, while the SCS global score shows excellent psychometric properties [38], recent research has challenged the validity of a total self-compassion score [54], proposing a two-factor model with two distinct concepts, self-compassion and self-coldness. Unfortunately, most of our studies used the SCS-SF, where the analysis of self-compassion on a subcomponent level is not recommended [39]. Thus, we could not delineate self-compassion from self-coldness in a subgroup analysis.

To conclude, this review supports (a) the association of self-compassion and self-reported sleep quality across diverse operationalizations and samples and (b) the causal role of self-compassion interventions in improving sleep quality. We would argue that in recent years, scientists have paid increasing attention to the replicability of results. We consider our mini meta-analysis as a step against questionable research practices such as p-hacking and file-drawer effects [55] by providing a transparent quantitative synthesis of our research program. These results contribute to a growing body of review literature on self-compassion. Future research should investigate moderating effects of intervention type, duration of intervention, and type of target population. 

## Figures and Tables

**Figure 1 behavsci-10-00064-f001:**
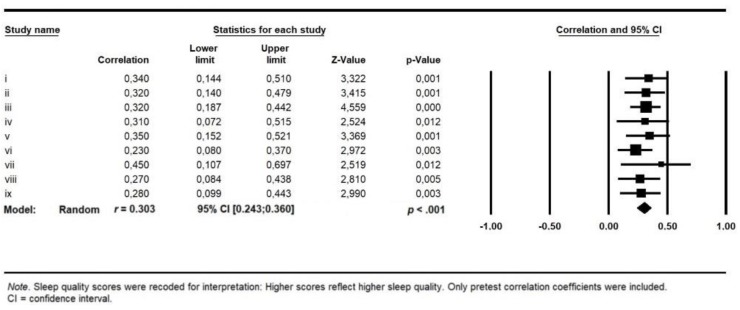
Meta-analysis results for SC-SQ-corrections of nine studies.

**Figure 2 behavsci-10-00064-f002:**
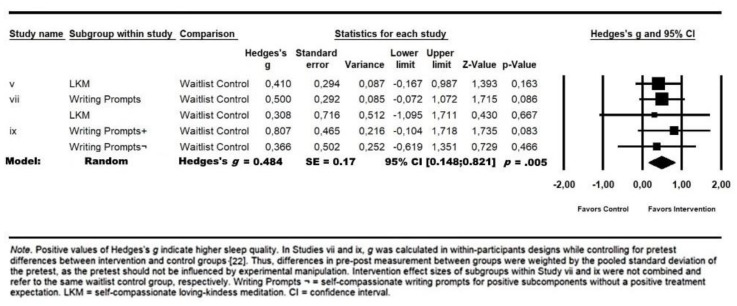
Meta-analysis results for SC-intervention effects on SQ of three studies.

**Table 1 behavsci-10-00064-t001:** Characteristics of studies examining the (causal) relationship between self-compassion and sleep quality. NR: not reported.

First Author and Year	Notation	Publication Status	Sample (*N*)	Mean Age (*SD*)	Gender (% Female)	Design	Scales/Interventions Used
Kombeiz and Stahlberg (2012)	i	Unpublished diploma thesis	University: students and community (*N* = 91)	27.08 (7.94)	77.2%	Correlational	SCS [12]; Sleep Quality Index [17]
Popova, et al. (2014)	ii	Unpublished diploma thesis	University: students and community (*N* = 109)	25.90 (6.19)	69.2%	Correlational; diary study (one week)	SCS [12]; Sleep Quality Index [17]
Stahlberg et al. (2016)	iii	Unpublished research project	University: students and community (*N* = 192)	27.00 (NR)	76.0%	Correlational	SCS [12]; Sleep Quality Index [17]
Butz and Stahlberg (2016) I	iv	Published Butz and Stahlberg (2018)	University: students (*N* = 65)	21.41 (5.65)	75.0%	Quasi-experimental (randomized order)	SCS-SF [13]; ISI [19]
Butz and Stahlberg (2016) II	v	Published Butz and Stahlberg (2018)	University: students (*N* = 88)	22.59 (3.43)	54.5%	Experimental; one-time intervention vs. waitlist	SCS [12]; Sleep Quality Index [17]/20min LKM vs. writing prompts
Tartter and Butz (2016)	vi	Unpublished bachelor’s thesis	University: students and community (*N* = 164)	27.50 (11.51)	68.0%	Correlational	SCS-SF [13]; Sleep Quality Index [17]
Butz and Stahlberg (2017)	vii	Published Butz and Stahlberg (2018)	Clinical: major depression episode (*N* = 30)	42.45 (11.54)	50.0%	Experimental; Pre–post intervention (20min 1 session, 5min 4 sessions) vs. waitlist	SCS-SF [13]; ISI [19]/20min LKM; 5min SC-break
Kuhn and Butz (2017)	viii	Unpublished bachelor’s thesis	University: students and community (*N* = 106)	40.00 (NR)	66.0%	Correlational	SCS-SF [13]; PSQI [18]
Butz and Stahlberg (2018)	ix	Published in partial fulfillment of the requirements for the degree of Doctor of Social Sciences	University: students (*N* = 111)	21.24 (3.18)	88.3%	Experimental; pre–post intervention (20min 1 session, 5min 6 sessions) vs. Waitlist	SCS [12]; Sleep Quality Index [17]/20min writing prompts; 5min SC-break

Note: Outcomes and scales reported in this table were the outcomes and scales included in the current analyses. All studies used additional scales beyond the scope of this meta-analysis and are not reported here. If necessary, all scales and interventions reported in this table were translated to German. Scale and intervention acronyms are as follows (listed alphabetically): ISI, Insomnia Severity Index; LKM, loving-kindness meditation; SCS, Self-Compassion Scale; SCS-SF, Self-Compassion Scale short-form.

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
