# Peer review of "The Relationship between Self-Compassion and Sleep Quality: An Overview of a Seven-Year German Research Program"

_behavsci, 2020, doi:10.3390/bs10030064_

Round 1
Reviewer 1 Report
In this work, Butz and Stahlberg provide an overview of their recent, single-center experience about the effect of self-compassion on subjective sleep quality. The manuscript is carefully written and the topic may be of interest, as this approach, if its efficacy is proven, might become a supplement to cognitive-behavioral therapy.
Here some concerns I would like to raise.
It is crucial that the authors specify that they are talking about subjective sleep quality. Also, although the authors analyzed the data consistently with the approach of a meta-analysis, given some severe limitations, this work cannot live up the standards of a meta-analyis, or a review. Indeed, the validity of unpublished works, in the absence of a peer-review process, is not fully certifiable; the heterogeneity of the populations (i.e. age, comorbidities) and the study designs might excessively confound the outcomes. Therefore, I would recommend the authors to edit the title and the text, adopting the wording they used at the end of the introduction (e.g. “an overview of a 7-year german research program"). In the introduction, I think the authors are mixing up too many concepts (i.e. insomnia symptoms, acute insomnia, chronic insomnia, paradoxical insomnia, sleep deprivation, etc) under the name of poor sleep without providing enough details about them. The clinical pathway leading to all of the aforementioned disorders is different and so are their consequences. Considering that, again, the authors are talking about subjective sleep quality, I would revise the first paragraphs under a different light. When the authors describe the self-compassion construct for the first time, it is not clear whether: the aspects are part of the self-compassion conceptualization, or they are different items in a questionnaire to be administered, or it is a way to classify people based on the self-compassion idea. The central assumption of social mentality theory is a bit vague. First, physiological arousals are, by definition, not pathological, thus, decreasing their number would not beneficially impact objective (but also subjective) sleep quality. Second, sleep architecture in chronic insomniacs seems to not vary much from healthy controls. Third, it is not entirely clear the physiologic mechanism by which self-soothing should reduce arousals. Fourth, I am not aware of any disorder, other than PTSD and parasomnias (which are not related to this topic), that would increase the number of arousals in response to social threats. When the authors first mention the self-compassion interventions, it is not clear what they would be (I think this topic remains poorly explained along the entire manuscript). “Trait self-compassion was significantly related to self-reported overall 214 sleep quality across the nine studies, r = 0.303”. Could this be explained by the fact that more self-compassionate people also tend to look at their perceived sleep quality in a more "positive, optimistic" way? Please discuss.
Other minor concerns as follows:
The authors should list all the components/items of the self-compassions questionnaires. Also, I suppose a high score means a high level of self-compassion. Please explain further. Line 52: “due to” does not read right. Line 60: “a” to be deleted. Line 71: please provide examples of shorter mindfulness interventions. Also note that, as aforementioned, the self-compassion-based interventions would likely need a longer and more consistent duration to be more effective, which brings us to the same problems that CBTI has (e.g. time consuming). As I mentioned at the beginning of the review, maybe the approach based on self-compassion could be useful in a setting where it is supplementary/complimentary to the CBTI. Please discuss. Line 137: “experimental studies”. Only one is cited. Line 139: I suppose the authors meant “manuscripts”. Line 150: only one study is referred into the bibliography.
Reviewer 2 Report
In the present study, the authors have performed a meta-analysis of some of their published and newer unpublished data to show a correlation between self-compassion and sleep quality. The introduction and methods sections are very nicely explain with sufficient details and proper references. The discussion is also very nicely written and mentions the potential limitations as well.
Author Response
Thank you for your review.